# Combining sound with tongue stimulation for the treatment of tinnitus: a multi-site single-arm controlled pivotal trial

Michael Boedts [1,2], Andreas Buechner[3,4], S. Guan Khoo[5,6], Welmoed Gjaltema[7], Frederique Moreels[1], Anke Lesinski-Schiedat[3,4], Philipp Becker[4], Helen MacMahon[8], Lieke Vixseboxse[7], Razieh Taghavi[7], Hubert H. Lim [9,10,11] ✉ & Thomas Lenarz[3,4]

Bimodal neuromodulation is emerging as a nonsurgical treatment for tinnitus. Bimodal treatment combining sound therapy with electrical tongue stimulation using the Lenire device is evaluated in a controlled pivotal trial (TENT-A3, NCT05227365) consisting of 6-weeks of sound-only stimulation (Stage 1) followed by 6-weeks of bimodal treatment (Stage 2) with 112 participants serving as their own control. The primary endpoint compares the responder rate observed in Stage 2 versus Stage 1, where a responder exceeds 7 points in the Tinnitus Handicap Inventory. In participants with moderate or more severe tinnitus, there is a clinically superior performance of bimodal treatment (58.6%; 95% CI: 43.5%, 73.6%; $p = 0.022$) compared to sound therapy alone (43.2%; 95% CI: 29.7%, 57.8%), which is not observed in the full cohort across all severity groups. Consistent results are observed for the secondary endpoint based on the Tinnitus Functional Index (bimodal treatment: 45.5%; 95% CI: 31.7%, 59.9%; sound-only stimulation: 29.6%; 95% CI: 18.2%, 44.2%; $p = 0.010$), where a responder exceeds 13 points. There are no device related serious adverse events. These positive outcomes led to FDA De Novo approval of the Lenire device for tinnitus treatment.

Tinnitus is a phantom auditory sensation that is coded within the brain that affects 10–15% of the population[1–6], where approximately 6-11% of the tinnitus population experience bothersome tinnitus and 2–8% experience a severe form of tinnitus[3,7], with limited treatment options[8–10]. Encouragingly, there is growing evidence across multiple animal and human studies demonstrating that bimodal neuromodulation combining sound with electrical stimulation of peripheral nerves, such as trigeminal and somatosensory nerves, can drive significant neural plasticity relevant for tinnitus treatment and improve tinnitus symptoms[11–22].

One emerging bimodal treatment approach is combining sound with electrical stimulation of the tongue; neurophysiological data in animals demonstrated that one of the strongest drivers of brain plasticity within the auditory cortex and midbrain was achieved with electrical stimulation of the tongue compared to other body regions[19]. These positive findings were further supported by two large-scale

[1]BRAI3N Clinic, Gent, Belgium. [2]Maria Middelares General Hospital, Gent, Belgium. [3]Department of Otolaryngology, Hannover Medical School, Hannover, Germany. [4]German Hearing Center (DHZ), Hannover Medical School, Hannover, Germany. [5]St. James's Hospital, Dublin, Ireland. [6]St. Vincent's Hospital, Dublin, Ireland. [7]Avania, Bilthoven, The Netherlands. [8]Otologie Clinic, Dublin, Ireland. [9]Neuromod Devices Limited, Dublin, Ireland. [10]Department of Otolaryngology-Head & Neck Surgery, University of Minnesota, Minneapolis, MN, USA. [11]Department of Biomedical Engineering, University of Minnesota, Minneapolis, MN, USA. ✉e-mail: hlim@umn.edu

**Fig. 1 | Bimodal neuromodulation treatment, study design and participant flow diagram.** Image in panel (**a**) reprinted from "Conlon, B. et al. Different bimodal neuromodulation settings reduce tinnitus symptoms in a large randomized trial. *Sci. Rep.* **12**, 10845 (2022)". Copyright 2022 by Nature Portfolio. **a** Lenire bimodal neuromodulation device developed by Neuromod Devices (Dublin, Ireland). Sound stimulation is delivered through wireless headphones and electrical stimulation is presented to the anterior-dorsal surface of the tongue via a 32-site electrode array, which is coordinated by a battery-powered controller. **b** Design of the TENT-A3 study with 6 weeks of sound-only stimulation in Stage 1 followed by 6 weeks of bimodal treatment when tongue stimulation is added to sound-only stimulation in Stage 2. **c** Participant flow diagram. Intention-to-Treat (ITT) population consists of all participants who met the eligibility criteria (including a THI ≥ 38 at the screening visit), were enrolled in the investigation, and were fitted with the investigational device. The ITT full cohort (*n* = 112) consists of all participants from the enrollment visit, while the ITT moderate or worse severity cohort (*n* = 44) consists of participants who had a THI ≥ 38 at the 6-week interim visit. ITT was used for the primary endpoint analysis where missing values were imputed using Multiple Imputation (MI) described in the "Methods" section. Statistical analysis for the primary performance endpoint compares the responder rate in the second 6-week period of bimodal treatment to the responder rate in the first 6-week period of sound-only stimulation.

clinical trials across 517 enrolled tinnitus participants where a take-home device that was used for 12 weeks of treatment (Fig. 1a; Lenire® developed by Neuromod Devices, Ireland; minimum compliance of 36 hours) was able to clinically significantly reduce tinnitus symptoms and provide patient-reported benefit in more than two-thirds of study participants (TENT-A1 study, clinicaltrials.gov: NCT02669069; TENT-A2 study, clinicaltrials.gov: NCT03530306[12,13,23,24];). The study outcomes were based on two widely used and validated outcome instruments, Tinnitus Handicap Inventory (THI) and Tinnitus Functional Index (TFI)[25–28], as well as several exit interview questions.

The TENT-A1 and TENT-A2 studies were parameter optimization studies intended to hone in on the key stimulus features driving tinnitus benefits[12,13]. They provided consistent and strong clinical evidence in a large sample size that several different bimodal stimulation settings could reduce tinnitus symptoms, which was primarily driven by the combination of pure tones and tongue stimulation. The inclusion of different background noise components or varying spatiotemporal patterns of stimulation on the tongue relative to the sound components were not critical drivers in improving tinnitus symptoms across patients. Based on the findings from these two studies, a third large-scale confirmatory clinical trial was then performed to demonstrate the safety and efficacy of the Lenire bimodal treatment for tinnitus and to seek FDA approval (TENT-A3; clinicaltrials.gov: NCT05227365). Together with FDA guidance, we designed and executed a controlled pivotal clinical trial based on the findings from the TENT-A2 study. In particular, TENT-A2 suggested the criticality of both the pure tones and tongue stimulation components of bimodal treatment; thus, TENT-A3 was designed to confirm that adding tongue stimulation to sound-only stimulation drove additional clinically significant improvements in tinnitus symptoms. The study design of TENT-A3 consisted of 6 weeks of sound-only stimulation (i.e., different

pure tones presented sequentially over time) in Stage 1 followed by 6 weeks of bimodal treatment when tongue stimulation is added to sound-only stimulation in Stage 2 (Fig. 1b). The primary outcome measure was based on THI and compared the responder rate observed in Stage 2 versus Stage 1, where a responder is defined as a participant with a clinical improvement in THI of at least 7 points within the corresponding treatment stage based on multiple studies supporting the clinical significance of 7 points[27,28]. The FDA recommended a prospective, single-arm, repeated measures study design where each participant served as their own control, and with a criterion in which participants had to achieve a clinically significant improvement in tinnitus symptoms with bimodal treatment during Stage 2 above and beyond what was already achieved with sound therapy during Stage 1. Thus, this rigorous study design required demonstration that bimodal treatment achieves additional or synergistic therapeutic contribution with the tongue stimulation component that exceeds what is achievable with sound therapy alone. A sham controlled study was not possible because both sound and tongue components involve suprathreshold stimuli that participants are expecting during treatment, and thus the participants would know if they received a sham condition. Based on further feedback from the FDA, the primary endpoint analysis was performed for the full cohort of participants, as well as for specific THI severity groups. These efficacy analyses together with an acceptable safety profile of the Lenire bimodal treatment led to FDA De Novo approval on March 6, 2023 (DEN210033; creation of a new medical device regulation name and code: Combined acoustic and electrical external stimulation device for the relief of tinnitus, QVN).

The TENT-A3 study enrolled 112 tinnitus participants across three separate clinical sites (BRAI3N, Belgium, PI: Michael Boedts; German Hearing Center Hannover, Germany, PI: Andreas Buechner and

Thomas Lenarz; St. James's Hospital, Ireland, PI: Guan Khoo). The study protocol was independently reviewed and approved by the Research Ethics Committee of Universitair Ziekenhuis Antwerpen in Belgium (BUN B3002021000174), Research Ethics Committee of Medizinische Hochschule Hannover in Germany (10199_BO_S_2022), and the National Office for Research Ethics Committees in Ireland (22-NREC-MD-005). The study was registered at clinicaltrials.gov (NCT05227365). All methods were carried out in accordance with relevant guidelines and regulations. Based on previous clinical trial results showing positive efficacy outcomes with Lenire bimodal treatment with enrolled participants who were at least sufficiently bothered by their tinnitus (inclusion criterion of THI ≥ 38, moderate or worse tinnitus[12]), those with a THI score of at least 38 were also enrolled into the TENT-A3 study (Supplementary Fig. 1). Participants were also required to have no more than a maximum hearing loss of 40 dB HL in the measurement frequencies in the range of 250 Hz to 1 kHz or of 80 dB HL in the measurement frequencies in the range of 2 to 8 kHz either unilaterally or bilaterally. The participants were instructed to use the Lenire device for one hour per day for 12 weeks, where the first 6 weeks consisted of sound-only stimulation and the second 6 weeks consisted of tongue stimulation added to sound-only stimulation. Primary endpoint analyses based on the THI are presented in this paper. Additional secondary analyses based on the TFI and exploratory analysis based on the Health Utilities Index Mark III (HUI3), as well as two satisfaction questions and safety outcomes are also presented in this paper.

## Results

### Characteristics and summary of study participants

Participants were recruited through various methods at the investigational sites, including advertising for the trial on an online forum and radio advertisements. During pre-screening, 2877 potential participants filled out an online form (Fig. 1c). Of the 2877 candidates, 2655 of them were excluded and 222 individuals were screened in clinic. Of these potential participants, 112 were enrolled based on inclusion and exclusion criteria (screen failure reasons provided in Supplementary Tables 1–3); they then signed an informed consent document and were fitted with the Lenire device at the enrollment visit. The study was completed on October 25, 2022. One participant was lost to follow-up during Stage 1, while six were lost to follow-up during Stage 2, resulting in 105 participants for which all relevant data were available for analysis. Missing values were imputed using Multiple Imputation (MI), as described in the "Methods" section, to perform the primary endpoint analyses using the Intention-to-Treat (ITT) population, as listed in Fig. 1c.

As shown in Fig. 1c, there was a high retention rate in the study, with 111 out of the 112 participants attending the interim visit (99.1% retention rate), and 105 out of the 112 participants attending the final visit (93.8%). The high retention rate is consistent with the high treatment compliance rate of 92.0% in Stage 1 and 82.4% in Stage 2 (Supplementary Fig. 2). The Lenire device logged the daily usage of each participant. Compliance in this study was defined as device usage of at least 18 hours in Stage 1 and at least 18 hours in Stage 2. Hearing thresholds across all enrolled participants are shown in Supplementary Fig. 3; the Lenire device fitting process requires hearing threshold values to be inputted into the software to ensure the sound stimulation component is audible for each participant.

In terms of sex, there were more males than females enrolled in the study (77 versus 35 participants, respectively; Table 1), which aligns with supporting literature where the prevalence of tinnitus is higher in males than females across demographic groups[5,29,30] and males are more likely than females to discuss tinnitus with a healthcare provider[31]. Furthermore, real-world evidence presented later in this paper shows that those seeking Lenire treatment consist of more males than females, which is relevant for FDA approval and the generalizability of treatment to the tinnitus population seeking clinical care. Other characteristics of enrolled participants at screening are presented in Table 1.

### Bimodal treatment achieves clinical efficacy for moderate or worse tinnitus

The primary endpoint agreed upon with FDA comprises a responder rate analysis comparing the rate in Stage 2 versus the rate in Stage 1, where a responder is defined as a participant with an improvement in THI score of at least 7 points within the corresponding treatment stage. When completing the primary endpoint analysis for the full cohort, the responder rate during Stage 2 (sound and tongue stimulation) was 43.3% ± 4.8% (95% CI: 33.9% to 52.7%) compared to the responder rate during Stage 1 (sound-only stimulation) of 63.3% ± 4.6% (95% CI: 54.3% to 72.2%; Table 2). The 43.3% response during Stage 2 is the additional clinically significant improvement in tinnitus symptoms (at least 7 more points in THI) when adding tongue stimulation to sound-only stimulation above what was already achieved during Stage 1 with sound-only stimulation. In other words, during Stage 1, there were 63.3% of participants who obtained at least 7 points improvement in THI in response to sound-only stimulation, with 36.7% of participants obtaining less than 7 points improvement; then in Stage 2, these same participants had to also achieve at least 7 more points of improvement above what they already achieved in Stage 1 to be considered a responder in Stage 2, which was a rigorously high criterion for success. Although 43.3% is an encouraging outcome for tinnitus treatment with bimodal stimulation, the primary endpoint of demonstrating a greater responder rate in Stage 2 beyond what is observed in Stage 1 was not achieved. Even though the study recruited participants who are sufficiently bothered by their tinnitus (inclusion criterion of THI ≥ 38 for moderate or worse tinnitus[26,27]), the high responder rate of 63.3% to sound therapy in Stage 1 resulted in many participants no longer or only minimally being bothered by their tinnitus at the start of bimodal treatment in Stage 2. To account for this floor effect and to address FDA's feedback for identifying participants with specific severity levels who can benefit from bimodal treatment, the primary analysis was also performed in the group of participants who still showed a moderate to catastrophic severity of tinnitus handicap when starting bimodal treatment (i.e., THI score ≥ 38 at the interim visit; n = 44). The primary endpoint analysis for this moderate or worse severity group showed a clinically significant superior performance of the Lenire device with combined sound and tongue stimulation (58.6% ± 7.7%; 95% CI: 43.5% to 73.6%) when compared to sound-only stimulation (43.2% ± 7.5%; 95% CI: 29.7% to 57.8%; p = 0.022; Table 2). For completeness, the responder rate over the full 12 weeks of treatment is also shown in Table 2, corresponding to 79.4% for the full cohort. A similar response of 76.0% was observed over the 12 weeks of treatment in participants who were at least moderately bothered by their tinnitus when starting bimodal

**Table 1 | Characteristics of enrolled participants**

| Characteristics | Units | Full cohort |
|---|---|---|
| Total enrolled | Number of participants | 112 |
| Sex: male | Number of participants [% enrolled] | 77 [68.8%] |
| Sex: female | Number of participants [% enrolled] | 35 [31.3%] |
| Age at screening | Years (mean [SD]) | 48.9 [12.6] |
| Tinnitus duration at screening | Years (mean [SD]) | 4.3 [3.1] |
| THI at screening | Points (mean [SD]) | 50.1 [11.3] |
| Mean hearing loss at screening | dB HL (mean [SD]) | 17.7 [10.5] |

Mean hearing loss is the average of hearing thresholds across frequencies of 0.25, 0.5, 1, 2, 3, 4, 6, and 8 kHz for both ears. dB HL: decibel hearing level; THI: Tinnitus Handicap Inventory.

**Table 2 | Primary endpoint analysis for bimodal treatment for tinnitus based on the Tinnitus Handicap Inventory (THI)**

| ITT population - all severity group (n = 112) | | | |
|---|---|---|---|
| | Sound therapy (Stage 1) | Addition of tongue stimulation to sound therapy (Stage 2) | Full 12 weeks of treatment (Stages 1 and 2) |
| Estimates ± SE | 63.3% ± 4.6% | 43.3% ± 4.8% | 79.4% ± 4.0% |
| 95% CI | 54.3%, 72.2% | 33.9%, 52.7% | 71.6%, 87.2% |
| ITT population - moderate or worse severity group (THI ≥ 38, n = 44) | | | |
| | Sound therapy (Stage 1) | Addition of tongue stimulation to sound therapy (Stage 2) | Full 12 weeks of treatment (Stages 1 and 2) |
| Estimates ± SE | 43.2% ± 7.5% | 58.6% ± 7.7% | 76.0% ± 6.8% |
| 95% CI | 29.7%, 57.8% | 43.5%, 73.6% | 62.7%, 89.2% |

Primary endpoint analysis comparing the responder rate observed during Stage 2 (the second 6-week period of treatment from interim visit to final visit) to the responder rate observed during Stage 1 (the first 6 weeks of treatment from enrollment visit to interim visit), where a responder is defined as a participant with an improvement in THI score of at least 7 points within the corresponding treatment stage. The responder rate ± SE and corresponding 95% CI at the different stages for the Intention-to-Treat (ITT) populations for the full cohort and for the moderate or worse severity group (THI ≥ 38) are presented in the table. Statistical analysis for the primary endpoint is described in detail in the "Methods" section. The stratified primary endpoint analysis in the moderate or worse severity group showed a clinically significant superior performance of the Lenire device with combined sound and tongue stimulation (58.6% ± 7.7%) when compared to sound-only stimulation (43.2% ± 7.5%; $p$ = 0.022; based on the THI severity grouping and statistical analysis accepted by the FDA for De Novo regulatory approval in the United States).

treatment. Clinical efficacy results shown in Table 2 are also consistently observed for male and female cohorts, as shown in Supplementary Table 4, 5.

In addition to THI, analyses for a secondary endpoint measure based on TFI were also performed in the study, as shown in Supplementary Table 6. Consistent with THI, the analysis for TFI showed that the moderate or worse severity group showed a clinically significant superior performance of the Lenire device with combined sound and tongue stimulation (45.5% ± 7.5%; 95% CI: 31.7% to 59.9%) when compared to sound-only stimulation (29.6% ± 6.9%; 95% CI: 18.2% to 44.2%; $p$ = 0.010), where a responder is defined as a participant with an improvement in TFI score of at least 13 points.

As part of the study design, FDA recommended analysis based on specific THI severity groups. The standard THI categories consist of five severity levels of tinnitus handicap: none/slight (0–16), mild (18–36), moderate (38–56), severe (58–76) and catastrophic (78–100). To identify an appropriate range of THI severity levels relevant for effective bimodal treatment that could be applied to the TENT-A3 efficacy analysis, real-world evidence (RWE) was obtained and analyzed as supplementary data (Supplementary Table 7 and Supplementary Fig. 4). The RWE data was collected from a single-site, observational retrospective chart review study performed at the Otologie Clinic based in the Hermitage Medical Clinic (Dublin, Ireland). Patients were fitted with the Lenire device, and data at the 6-week clinical visit was available for analysis. A summary of the methods for the RWE data is provided in the Supplementary Information together with Supplementary Table 7 and Supplementary Fig. 4. The RWE shows that participants with a THI score below 38 generally exhibit small to no improvements in tinnitus symptoms with bimodal stimulation using the Lenire device, whereas significantly greater improvements are achieved for those with a THI score greater than or equal to 38 when starting bimodal treatment. Based on the RWE results, the primary endpoint for the TENT-A3 study was analyzed by excluding those individuals from the none/slight and mild categories (THI < 38) at the interim visit to focus on participants who were still sufficiently bothered by their tinnitus when starting bimodal treatment. This THI severity grouping analysis reveals a clinically significant superior performance of the Lenire device with combined sound and tongue stimulation when compared to sound-only stimulation for those who are at least moderately bothered by their tinnitus (Table 2). The individual data for each of these participants are shown in Fig. 2, in which the total cumulative change in THI score across the full 12-week treatment period is plotted in terms of the contributions from Stage 1 and Stage 2. For each participant, represented as a bar, the net contribution of Stage 2 (i.e., the addition of tongue stimulation to sound-only stimulation) is represented in green, demonstrating that there is an additional reduction in THI scores that can be attributed to tongue stimulation

across the majority of participants (i.e., green bars below the abscissa).

Consistent with Table 2 and Fig. 2 in supporting the additional benefit of improving tinnitus symptoms by adding tongue stimulation to sound-only stimulation, Supplementary Table 8 further shows that bimodal treatment can convert non-responders to responders of treatment in the full cohort of participants. 64.9% of participants who did not clinically improve with sound-only stimulation during Stage 1 of the study exhibited clinically meaningful improvements in tinnitus symptoms in response to bimodal stimulation during Stage 2. This finding is relevant for real-world situations where some tinnitus patients who have already used sound therapy approaches could then seek bimodal treatment for additional clinically meaningful improvements in tinnitus symptoms.

## High satisfaction and acceptability of Lenire treatment

At the final visit, participants completed two satisfaction or acceptability questions, in addition to assessing clinical efficacy with the THI used for the primary endpoint analysis. One question showed that 62.9% (66/105) of the participants experienced benefit from bimodal treatment with combined sound and tongue stimulation during Stage 2 of the study (Fig. 3a). Another question showed that 88.6% (93/105) of the participants would recommend the Lenire treatment to others with tinnitus (Fig. 3b). Interestingly, there was a higher rate of cases for recommendation than for those who indicated benefiting from bimodal treatment, which further supports the high acceptability of the Lenire device such that participants were willing to refer it to others suffering from tinnitus even if it was not guaranteed that they would benefit from the treatment. These high satisfaction and acceptability rates are consistent with the high treatment compliance rate of 82.4% for bimodal treatment (Supplementary Fig. 2).

In addition to satisfaction questions, a quality-of-life exploratory assessment based on HUI3 was performed in the study. The original intention of including the HUI3 was to determine if it could be sensitive enough to track overall health-related quality of life in tinnitus patients, which will be important for later seeking reimbursement for a tinnitus intervention. Yet, there was minimal change in the HUI3 score from the screening visit to the final visit (Supplementary Table 9). The HUI3 has general health questions pertaining to vision, ambulation, dexterity, emotion, cognition, pain, speech, and hearing. Because there is a hearing component in the HUI3, it was included in the TENT-A3 study. However, 78 out of 80 participants who completed the HUI3 already had the best score possible for the hearing category during the screening visit (note that scores range from 1 to 5 or 1 to 6 for each of the eight categories, with one being the best condition; these scores are then mapped to a total score for the HUI3 between 0 and 1 corresponding to dead and healthy, respectively). The total score of HUI3

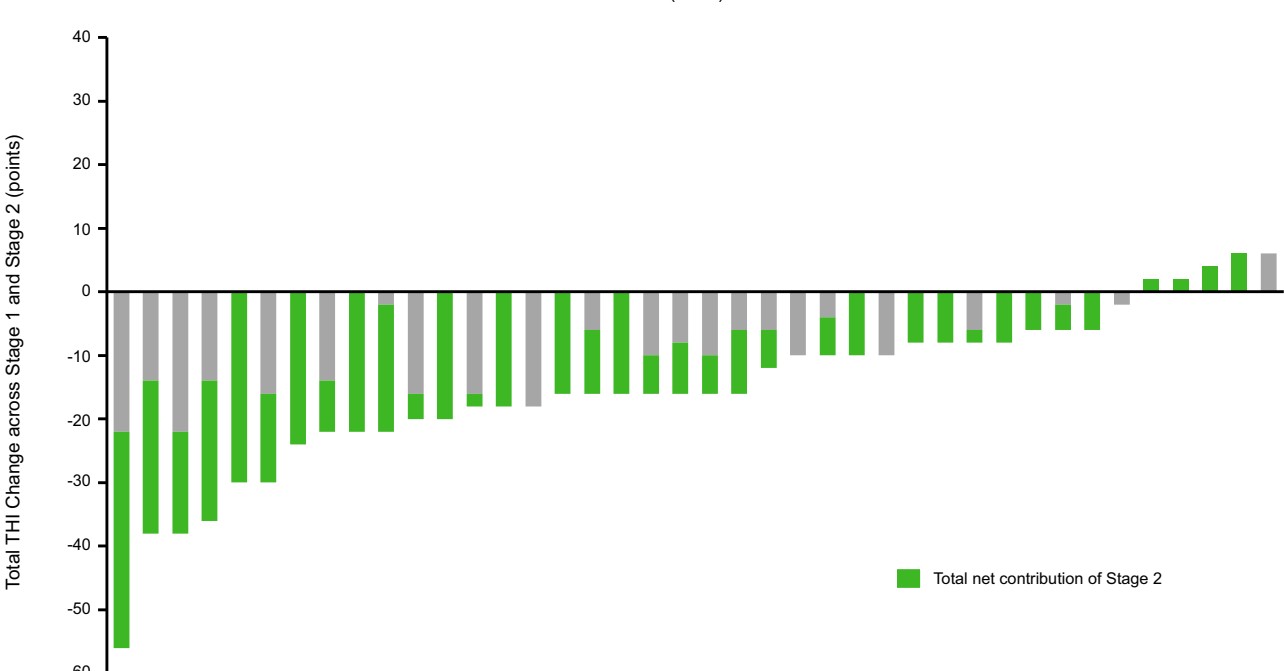

Total THI change across Stage 1 and Stage 2 ITT population - complete cases for moderate or worse severity group
THI ≥ 38 at interim (n=40)

**Fig. 2 | Total Tinnitus Handicap Inventory (THI) change across Stage 1 and Stage 2 for ITT population-complete cases for moderate or worse severity group (THI ≥ 38) at interim (n = 40).** Note that data is available for 40 participants representing the complete cases population displayed in this figure; for the primary endpoint analysis, missing values were imputed to obtain the ITT population (n = 44). Stage 1 consists of sound-only stimulation (gray bars), whereas tongue stimulation is added to sound-only stimulation in Stage 2 (green bars).

across participants was also high at the start of the study at 0.8, indicating study participants were generally healthy irrespective of their tinnitus. Therefore, the HUI3 did not prove to be an appropriate or sensitive enough assessment tool for tracking a tinnitus intervention.

**Acceptable safety profile of Lenire treatment**

In this study, there were no device-related serious adverse events (SAEs). The total number of potentially device related adverse events (AEs) are listed in Table 3. AEs and device deficiencies were documented and categorized in accordance with ISO14155:2020. These AEs were documented based on responses provided by the participants at the in-person visits, as well as from phone calls or emails between or after each visit, where the investigators closely tracked the AEs and their resolution during each stage of treatment or across the study. Each AE was categorized by type and seriousness according to the definitions provided in ISO14155. Whether an AE was related to the device or procedures was also distinguished. All available details for each AE were recorded in the participant CRFs (case report forms), and each AE was assessed by the site PI along with an independent medical monitor who categorized the AEs by their relationship to the investigational device in terms of seriousness, severity (mild, moderate, or severe), onset date, resolution status, any action taken, and if there were any sequelae. These AEs were then further grouped into three relatedness categories. The ISO14155 does not provide guidance on the causality assessment of AEs. For the causality assessment of all AEs, the MDCG 2020-10/1 guideline was followed. This guidance is specifically aimed at SAEs; however, it was extrapolated to all AEs for this study. In brief, causal relatedness was defined as an AE associated with the investigational device beyond reasonable doubt. Probably device related was defined as having a relationship with the use of the investigational device that seems relevant and/or the event cannot be reasonably explained by another cause. Possibly device related was

defined as having a relationship with the use of the investigational device that was weak but cannot be ruled out completely. Not device-related was defined as an event not having a temporal relationship with the device or not following a known response pattern to the device. The AEs were then further classified into mild, moderate, or severe categories. Mild severity AEs correspond to awareness of signs or symptoms but are easily tolerated and are of minor irritant type, causing no or minimal loss of time from normal activities; these symptoms are transient and do not require therapy or a medical evaluation. Moderate cases are events that introduce a low level of inconvenience or concern to the participant and may interfere with daily activities; moderate experiences may cause some interference with functioning. Severe cases are events that substantially interrupt the participant's normal daily activities and generally require systemic drug therapy or other treatment; these events are usually incapacitating.

During the bimodal treatment stage, there were 18 device related AEs that were all mild cases (Table 3). During the sound-only stimulation period, there were 44 device-related AEs, in which 42 were mild cases and two were moderate cases (one panic attack and one increased tinnitus case that were possibly and probably device-related, respectively). Although there were 12 cases of increased tinnitus AEs during Stage 2 with bimodal treatment, there were 40 cases of increased tinnitus AEs during Stage 1 with sound-only stimulation, supporting the cause of this tinnitus increase to be associated more with the sound component of treatment and/or the initiation of a new treatment paradigm. Overall, 96.8% (60/62) of the device-related AEs reported during the study were mild. Encouragingly, all AEs were resolved from the study, except one mild case that was lost to follow-up, demonstrating a very good safety profile for the Lenire treatment, in which nearly all AEs were mild and/or transient. The risks associated with the Lenire treatment were acceptable for obtaining FDA De Novo approval.

Overall, would you say you have benefitted from using this device with bimodal stimulation (during the second stage of treatment)?

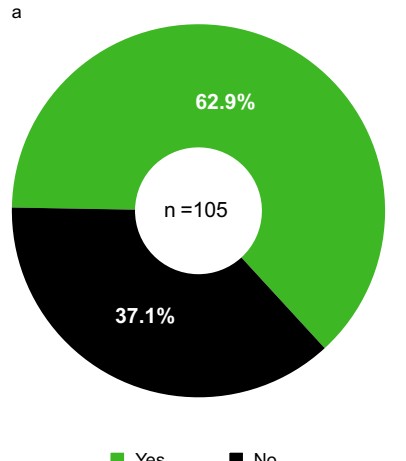

If you knew someone with tinnitus, would you recommend they try this treatment?

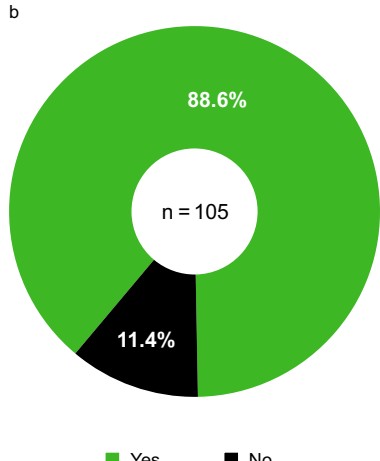

**Fig. 3 | Satisfaction or acceptability rate in using the Lenire treatment device.** Two questions relating to the participant's satisfaction (**a**, left) or acceptability (**b**, right) of the treatment device were asked at the final visit when the treatment ended, in which the percentage of yes or no responses are shown.

## Discussion

The controlled pivotal TENT-A3 trial (registered at clinicaltrials.gov: NCT05227365) was designed with guidance from FDA to confirm that adding tongue stimulation to sound-only stimulation achieves additional clinically significant improvements in tinnitus symptoms above what is possible with sound-only stimulation; thus, demonstrating the criticality of both the sound and tongue stimulation components in treating tinnitus with Lenire bimodal treatment. For individuals with a moderate to catastrophic severity of tinnitus handicap when starting bimodal treatment (i.e., THI score ≥ 38), a clinically significant superior

performance of the Lenire device with combined sound and tongue stimulation was achieved with 6 weeks of treatment (58.6%) when compared to sound-only stimulation (43.2%; $p = 0.022$; Table 2). This outcome occurred regardless of whether the participant benefitted or not from sound-only stimulation (Fig. 2). Furthermore, the responder rate over the full 12 weeks of treatment was 79.4% for the full cohort and 76.0% for those who were at least moderately bothered by their tinnitus when starting bimodal treatment. Consistent with the previous TENT-A1 and TENT-A2 studies[12,13], TENT-A3 also achieved high treatment compliance and satisfaction rates for bimodal treatment. At the final visit, 62.9% of participants indicated that they benefited from using the Lenire device with bimodal stimulation, which corresponds to the 6-week treatment period of Stage 2. Also, 88.6% said that if they knew someone with tinnitus, they would recommend they try this treatment. A high compliance rate of 82.4% was achieved with participants using bimodal treatment for at least the minimum compliance of 18 hours during Stage 2, with an average daily usage of 52.6 minutes for these treatment-compliant participants.

The positive clinical efficacy, compliance rates, and satisfaction outcomes for Lenire bimodal stimulation are further strengthened by the relative safety of the treatment. There were no treatment-related SAEs during the study, no device-related AEs that led to a withdrawal during the study, and no device deficiencies associated with an AE. Overall, 96.8% of the device-related AEs reported during the study were mild. Encouragingly, all AEs were resolved from the study, except one mild case that was lost to follow-up, demonstrating a very good safety profile for the Lenire treatment.

The positive clinical efficacy and safety outcomes for the Lenire bimodal treatment, which was further supported by consistent real-world data collected at the Otologie Clinic, led to FDA De Novo approval on March 6, 2023 (DEN210033). Recently published real-world data collected in Germany also showed consistent positive outcomes in tinnitus symptoms with Lenire bimodal treatment[32]. Lenire is approved by the FDA to provide bimodal stimulation for temporarily relieving the symptoms of tinnitus in patients 18 years of age and older suffering from at least moderate tinnitus severity. The novelty of the Lenire bimodal treatment supported by a favorable benefit-risk profile has led to the creation of a new medical device category and definition for tinnitus (Combined acoustic and electrical external stimulation device for the relief of tinnitus; product code: QVN; regulation number: 874.3410).

**Table 3 | Safety data recorded throughout the study**

| Number of device related mild adverse events (AEs) at Stage 1 and Stage 2 | | | | |
|---|---|---|---|---|
| | Stage 1 (sound-only stimulation) (n = 112) | | Stage 2 (bimodal stimulation) (n = 111) | |
| | Total number of events | Ongoing | Total number of events | Ongoing |
| Increased or worsening tinnitus | 40 | 0 | 12 | 1 |
| Glossodynia | 0 | 0 | 2 | 0 |
| Salivary hypersecretion | 0 | 0 | 2 | 0 |
| Dizziness | 1 | 0 | 1 | 0 |
| Headache | 1 | 0 | 0 | 0 |
| Dry throat | 0 | 0 | 1 | 0 |
| **Number of adverse events (AEs) by relatedness to device at Stage 1 and Stage 2** | | | | |
| Possibly device related | 17 | 0 | 10 | 0 |
| Probably device related | 7 | 0 | 2 | 1 |
| Causal relationship | 20 | 0 | 6 | 0 |

There were no treatment-related SAEs. The number of device-related mild AEs during Stage 1 and Stage 2 are shown at the top of the table. There were two moderate AEs not included in the upper half of the table that occurred during sound-only stimulation in Stage 1 (one panic attack case and one increased tinnitus case that were resolved; they were classified as possibly and probably device-related, respectively). Also listed in the lower half of the table are the number of AEs by relatedness during Stage 1 and Stage 2, which includes both mild and moderate AEs. In Stage 2, the ongoing increased tinnitus case was classified as probably device-related and lost to follow-up.

One limitation of the study is that the long-term benefits of bimodal stimulation with the Lenire device were not assessed after treatment ended due to time constraints posed by the FDA for providing clinical trial data in response to an initial De Novo submission. However, in the two previous large-scale TENT-A1 and TENT-A2 studies with Lenire bimodal treatment, participants were followed up for 12 months after treatment ended and exhibited a sustained efficacy effect that was observed in a large percentage of treatment-compliant participants[12,13]. Further support for the long-term clinical efficacy of bimodal treatment for tinnitus was recently demonstrated in a double-blind, crossover, single-center randomized clinical trial with a 3-month follow-up period[22]. In that study, 6 weeks of bimodal stimulation with combined sound and electrical stimulation of the face or neck region was implemented in tinnitus participants for 30 minutes per day, which showed clinically significant improvement in tinnitus symptoms that was greater than outcomes for sound-only stimulation, with benefits that could last for at least 12 weeks after treatment ended.

One consideration for interpreting the results of bimodal treatment is related to the design of the study. The Lenire bimodal treatment cannot be sufficiently blinded because both the sound and tongue stimuli are suprathreshold and noticeable, thus a sham control is not possible. Therefore, a high criterion for efficacy success was decided together with FDA, where we had to demonstrate not only that bimodal treatment is effective (e.g., over a sham control), but also that bimodal treatment is more effective than sound-only treatment. Furthermore, the design of the study required that additional improvements in tinnitus symptoms be achieved with bimodal stimulation in Stage 2 above and beyond the improvements already achieved in Stage 1 with sound-only stimulation. In other words, the success criterion requires more than a doubling of the response rate across Stage 1 through to Stage 2. We have not observed, to the best of our knowledge, appropriately designed sound therapy studies demonstrating a growing slope of improvement over time as treatment continues. Overall, our rigorously designed success criterion confirms that Lenire bimodal treatment achieves clinical efficacy for participants with moderate or worse tinnitus.

Another consideration for Lenire bimodal treatment is that clinical efficacy was demonstrated for individuals who have moderate or worse tinnitus symptoms (i.e., THI score ≥ 38). For those with mild or slight tinnitus symptoms (i.e., THI < 38), and considering the positive benefit of sound-only stimulation observed in Stage 1, sound therapy can be initially considered for these tinnitus patients, and if symptoms are not sufficiently addressed, then bimodal treatment may also be provided as a follow-on therapy. Encouragingly, 64.9% of participants who did not clinically improve in response to sound therapy during Stage 1 of the study exhibited clinically meaningful improvements in tinnitus symptoms in response to bimodal stimulation during Stage 2. Therefore, Lenire bimodal treatment offers a clinically validated approach that is now CE marked in Europe and FDA approved in the United States that can be used to treat tinnitus patients as a standalone approach or in combination with various sound therapies already available in hearing clinics.

## Methods

### Study design

TENT-A3 was a prospective, single-arm, repeated measures, multi-site investigation with the objective to determine whether the addition of tongue stimulation to sound-only stimulation provides an additional effect on tinnitus reduction when compared to sound therapy alone. The intervention was the Lenire bimodal treatment device developed by Neuromod Devices that delivers sound wirelessly via Bluetooth headphones while electrical stimulation is delivered to the surface of the tongue using a wired 32-sited electrode array (Fig. 1a). Due to the nature of the single-arm repeated measures investigation design, no

randomization or masking was applicable. The study was designed with guidance and feedback from the FDA to obtain appropriate data for FDA De Novo approval, where the effects of sound therapy were appropriately controlled to show the additional benefit of the addition of tongue stimulation to sound-only stimulation using the Lenire device.

In TENT-A3, 112 participants were enrolled in the study across three investigational sites in three different countries (Table 1). The study was led by independent investigators across these separate clinical sites. Recruitment was achieved through various channels, such as advertising on online forums or via online and radio advertisements. Eligible participants were screened at a screening visit after providing informed consent and were invited to an enrollment visit as depicted in Fig. 1c. At the enrollment visit, participants were trained and fitted with the Lenire device (CE-marked Class IIa; Neuromod Devices, Dublin, Ireland; Fig. 1a). Participants were recommended to administer treatment for up to one hour per day, in two 30-minute sessions every day, for the 12-week period while enrolled in the investigation. Minimum compliance is defined as at least 18 h of device usage for each of the two stages of treatment. Participants completed 6-weeks of the first treatment stage (Stage 1, unimodal: sound-only stimulation) in their own home before being re-assessed at the interim visit (Fig. 1b). The participants were then switched to the second treatment stage (Stage 2, bimodal: sound and tongue stimulation) and completed the second 6-weeks of treatment, before attending the final visit (Fig. 1b), at which point the investigational period ended. Participants received two phone calls during the investigation to encourage treatment compliance, where the first call occurred approximately three weeks after the enrollment visit, and the second call occurred approximately three weeks after the interim visit.

Similar to the previous TENT-A1 and TENT-A2 trials, the Lenire device delivered sound wirelessly via Bluetooth headphones while electrical stimulation was delivered to the surface of the tongue using a wired 32-sited electrode array (Fig. 1a). The participant's pure-tone audiometric threshold (250–8 kHz) was measured at the screening visit and subsequently used to configure the sound stimuli to be comfortably audible above their hearing threshold at each tone frequency. The participant could adjust the default sound stimulus loudness between −12 dB and +12 dB during treatment using volume buttons on the controller. For safety reasons, the upper stimulus was limited for participants commensurate with their degree of hearing loss.

Electrical tongue stimulation intensity was configured for each participant by adjusting the intensity from sub-threshold to suprathreshold sensations to a comfortable intensity across different electrodes. This intensity was set as the calibrated setting, and the participant could move up or down six steps from the default level, with each step changing the pulse width by 8%. The calibration process is designed to deliver the lowest amount of electrical stimulation needed to achieve a perceptible stimulus for each individual participant. The treatment device reverts to the default intensities at the start of each new session and all changes to stimulation settings are recorded on the device log.

In the first 6-weeks, sound-only stimulation (known as the PS6 setting without the tongue stimulation component) was delivered, with sequences of pure tone bursts presented binaurally. In the subsequent 6-weeks, bimodal stimulation (PS6) was delivered with the same sound stimulus from the first 6-weeks, with the addition of tongue stimulation paired with different tone bursts. Further details on PS6 features have been described previously in Conlon et al.[12].

AEs were documented and categorized by type and seriousness in accordance with ISO14155:2020. These AEs were then further grouped into three relatedness categories. The ISO14155:2020 does not provide guidance on the causality assessment of AEs. For the causality assessment of all AEs, the MDCG 2020-10/1 guideline was followed.

## Ethics approval

The study protocol was approved by the Research Ethics Committee of Universitair Ziekenhuis Antwerpen in Belgium (BUN B3002021000174), Research Ethics Committee of Medizinische Hochschule Hannover in Germany (10199_BO_S_2022), and the National Office for Research Ethics Committees in Ireland (22-NREC-MD-005). The study was registered at clinicaltrials.gov (NCT05227365). All methods were carried out in accordance with relevant guidelines and regulations. The study protocol is available in the Supplementary Information. The study participants provided written informed consent and were enrolled in the study during the period between March 21, 2022 and June 7, 2022. Participants received a small monetary contribution towards travel, parking, and food expenses for clinic visits. At the end of the study, the participants were provided the option to return or keep the treatment device. There were no potential self-selection bias or other biases apparent to the study investigators.

## Participants

The study recruited participants who were 18 years and over at the time of consent with subjective, chronic tinnitus (≥ 3 months and ≤ 10 years) with a THI score of greater than or equal to 38 points. An eligible participant had to be able to read and understand Dutch/Flemish/English or German (depending on the clinical site), was willing and able to provide and understand informed consent, and was willing to commit to the full duration of the investigation. Participants were required to have no more than a maximum hearing loss of 40 dB HL in the measurement frequencies in the range of 250 Hz to 1 kHz or of 80 dB HL in the measurement frequencies in the range of 2–8 kHz either unilaterally or bilaterally.

Potential participants were excluded if they had objective or pulsatile tinnitus, or showed abnormal otoscopy or tympanometry that may be contributing to or causing the tinnitus. Meniere's disease, temporomandibular joint disorder (TMJ), burning mouth syndrome (BMS), previously diagnosed with psychosis or schizophrenia, or hospitalization or visit to a physician for head or neck injury in the previous 12 months were also exclusion criteria. Further exclusions included commencement of usage of hearing aid within the last 90 days, pregnancy, oral piercings that cannot or will not be removed for the second stage of the investigation, neurological condition that may lead to seizures or loss of consciousness, Mini Mental State Examination (MMSE) score less than 20 (severe cognitive impairment), State-Trait Anxiety Inventory (STAI) score of greater than 120, any type of electro-active implanted device, initiated or ceased prescription medications or treatments in the previous three months that may impact the outcomes of the investigation, or being involved in relevant medico-legal cases. Candidates were also excluded if they had previously used Lenire, had previously been involved in a clinical investigation for tinnitus, or had an experimental or surgical treatment for tinnitus; or if the site principal investigator deemed the candidate to be unsuitable for the investigation for other reasons not listed above.

## Clinical study endpoints

The primary endpoint of TENT-A3 was the responder rate during the second 6-week period of treatment comprising combined sound and tongue stimulation compared to the point estimate of the responder rate observed during the first 6-week period of treatment comprising sound-only stimulation. A responder was defined as a participant with an improvement in THI score of at least 7 points[28]. The primary endpoint analysis was also conducted for specific THI severity groups, particularly for participants with moderate or more severe tinnitus severity (THI ≥ 38; includes moderate, severe, and catastrophic THI severity groups) when starting bimodal treatment.

The THI is a clinical outcome measure commonly used to assess tinnitus symptom severity[27,28,33]. The THI predominantly assesses the emotional and functional impact of tinnitus, in which 25 items are scored 4/2/0 on a categorical scale corresponding to yes/sometimes/no. The global score of the THI (i.e., sum of points across all 25 items) has a value from 0 to 100, with a higher score indicating a greater negative impact of tinnitus. The THI scores can also be categorized into five standard severity levels of tinnitus handicap: none/slight (0-16), mild (18-36), moderate (38-56), severe (58−76) and catastrophic (78−100)[33]. The MCID reported for THI is 7 points and represents a clinically meaningful change in tinnitus symptoms.

As described by McCombe et al.[33], tinnitus patients in the moderate THI severity group and above (THI ≥ 38) have tinnitus that interferes with their ability to carry out normal daily activities and often encounters a disturbed sleep pattern that can be associated with emotional distress, mood disorders, somatic pain, stress responsivity, and reduced quality of life. In comparison, patients in the none/slight and mild groups (THI < 38) experience but are not generally troubled by their tinnitus. The tinnitus is often easily masked by environmental sounds and easily forgotten with activities. The tinnitus may occasionally interfere with sleep but not daily activities. Based on these severity categories, it is anticipated that patients within the moderate group and above (THI of 38 to 100) are more bothered by their tinnitus and would be more likely to seek treatment at clinics. Similar to the TENT-A2 study and to be able to assess clinical efficacy in participants with bothersome enough tinnitus with room for improvement with an intervention, an inclusion criterion requiring participants to have a THI ≥ 38 at the screening visit was also implemented in the TENT-A3 study.

In addition to THI, two additional endpoint measures were included in the study: TFI and HUI3. The TFI was used as a secondary endpoint measure and the HUI3 was included as an exploratory endpoint measure for the study. The TFI assesses a range of tinnitus-related functional complaints experienced over the past week prior to assessment. Each of the 25 items is assessed on an 11-point Likert scale, and the sum of the scores is normalized to give a global score from 0 to 100, with a higher score indicating a greater negative impact of tinnitus[25,26]. The MCID reported for TFI is 13 points and represents a clinically meaningful change in tinnitus symptoms. The HUI3 is a quality-of-life assessment with 40 questions and provides descriptive evidence on multiple dimensions of health status, a score for each dimension of health, and a health-related quality-of-life score for overall health[34,35]. The utility scores for each dimension have interval-scale properties. The HUI3 total score ranges from 0.0 (considered dead) to 1.0 (healthy)[34,35].

## Statistical analyses

The primary endpoint analysis was developed together with the CRO Avania and with guidance from the FDA that was viewed as appropriate for our study design and Lenire treatment considerations. DFdiscover was used for data collection and management for the study. The primary endpoint analysis consisted of a single sample, one-sided normal approximation test (Z-test) for a binomial proportion with a significance level of 0.025 in order to test the responder rate in the second 6-weeks, attributed to the addition of tongue stimulation to sound-only stimulation, compared to the point-estimate of the responder rate in the first 6-weeks for sound-only stimulation. A one-sided test was based on two previous large-scale studies (TENT-A1 and TENT-A2 studies[12,13]), supporting that one direction of effect is clinically relevant, in which the significance level was appropriately adjusted to 0.025 in lieu of using a significance level of 0.05 for a two-sided test. The ITT population with imputed missing values was used as the primary analysis population. Missing data was handled using a Markov chain Monte Carlo multiple imputation[36,37]. For this method, 50 multiple imputed datasets were first generated to fill in all missing data for the predictors. Subsequently, separate imputation processes estimating the responder rate at each visit (interim visit and final visit) were

generated using the fully conditional specification (FCS) linear regression method. Inferences for the primary endpoint were evaluated on each of the 50 imputed data sets and results combined to yield the estimates, confidence intervals, and associated significance values. Statistical analyses were performed by Avania using SAS software version 9.4 or later and R version 4.1.2 or later.

For the sample size calculation of the study, the estimated responder rate of 45% for Stage 1 was based on relevant data from the previous TENT-A2 clinical trial and accounts for a reasonable upper bound for the placebo effect as observed in the literature[12,38]. The estimated responder rate for Stage 2 was based on relevant data from the previous TENT-A2 clinical trial, and using modified Wald binomial probabilities with 90% confidence leads to a required estimated responder rate of at least 61%, which was then rounded to 60% to account for a worst-case scenario responder rate. The power for sample size was (1-β) equal to 0.8 with a type 1 error rate (α) equal to 0.025. These specifications yielded a sample size estimate of 89 participants for the study. The sample size was increased to 112 to consider a 20% drop-out or attrition during the clinical investigation (i.e., 80% of 112 equals 89.6), including to accommodate the COVID-19 pandemic during the study. For the analysis of the cohort with a THI severity greater than or equal to 38 at the interim visit, and considering the conservative rule of thumb of np and n(1-p) greater than 10 at each treatment stage[39], 44 subjects in the cohort would be a large enough sample size to assume an approximately normal distribution for the difference of proportions; hence, justifying using a z-test for the hypothesis test, where p corresponds to the responder rate. The presented clinical trial results are also supported by the RWE data from 204 participants presented in the Supplementary Information, where the severity analysis was repeated on a cohort that had completed 6 weeks of bimodal neuromodulation and the RWE results are consistent with the TENT-A3 results.

At the screening visit, participants self-reported if they were male or female. Primary endpoint analyses for the ITT population for the full cohort and for the moderate or worse severity group were additionally carried out according to participants' self-reported sex. Of the 112 enrolled participants, 77 were male and 35 were female.

### Reporting summary

Further information on research design is available in the Nature Portfolio Reporting Summary linked to this article.

## Data availability

All relevant data associated with the published study are present in the paper or the Supplementary Information. Source data for the figures are also provided in this paper. Data related to the primary results, as presented in the paper, are available under restricted access as ethical approval is required as additional processing or analysis by third parties not involved in the clinical study was not covered in the approved protocol or patient consent. Access can be obtained, contingent on appropriate ethics approval and data sharing agreements, by contacting HHL (tent-admin@tinnitustrial.ie) for the purposes of confirming the analysis in the paper. Responses to valid requests will be reasonably attempted and initiated within 10 working days of receipt, beginning 3 months and ending 5 years after this article's publication. The raw individual-level participant data are not available due to data protection regulations in Europe and since the informed consent form signed by participants does not allow for sharing individual-level participant data to third parties outside of the scope of the study.

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

## Acknowledgements

We would like to thank J. Ost, J. Van Herk, E. Willis and the team at BRAI3N (Belgium clinical site); C. Kelly, L. Hartzel, C. Lawless, D. Reidy and the team at the Wellcome-HRB Clinical Research Facility in Dublin (Ireland clinical site); and P. Weinstein, J. Dima and the team at DHZ (Germany clinical site) for performing evaluations with the participants, organization/execution of the study, and data collection during the study. We also thank the Avania Clinical Research Organization (Bilthoven, Netherlands), to whom investigation-specific activities and responsibilities were delegated by the study sponsor, for expert guidance and support in executing the study, training the teams, overall study management, on-site and remote monitoring, data validation, statistical analysis, and data visualization. In particular, we would like to thank S. Roeles, M. Hermse, and S. Boukes for their significant contribution to the management, monitoring, and interpretation of the study, together with the other co-authors of this paper from Avania. We would also like to thank the independent medical monitor assigned by Avania, G. DuQuella, for reviewing, categorizing, and classifying the safety data. For the real-world evidence data collection and compilation, we would like to thank A. Sayers and her team at the Otologie clinic (Dublin, Ireland) for providing the data to add in the Supplementary Information. We further appreciate the technical assistance, device training, and administrative support provided by the team at Neuromod Devices (study sponsor), especially E. Meade and S.L. Leong. The study sponsor also managed communication with the FDA on the study design and analysis prior to the execution of the study and during the review stages for FDA De Novo approval. Finally, we thank Deborah Hall, Sven Vanneste, and Berthold Langguth for providing feedback on the design of the study and outcomes. The study was funded by Neuromod Devices.

## Author contributions

Conceptualization and methodology: M.B., A.B., S.G.K., W.G., A.L., H.H.L., and T.L. Investigation: M.B., A.B., S.G.K., F.M., A.L., P.B., H.M.M., and T.L. Project administration: W.G., H.H.L., and T.L. Data validation and analysis: L.V. and R.T. Writing and editing: M.B., A.B., S.G.K., W.G., F.M., A.L., P.B., H.M.M., L.V., R.T., H.H.L., and T.L.

## Competing interests

Authors declare that they have no competing interests, except HHL who is a consultant with financial interests for Neuromod Devices.
