## [Peer Review File · Nature Communications]

Combining sound with tongue stimulation for the treatment of tinnitus: a controlled pivotal trialREVIEWER COMMENTS

Reviewer #1 (Remarks to the Author):

This manuscript describes a follow up study of a bimodal stimulation device, the Lenire, designed to decrease tinnitus symptoms. The manuscript is very well written. Many thanks to the authors for a very interesting study. The results are noteworthy in that a significant proportion of the participants demonstrated reductions in self-perceived tinnitus handicap following 6 weeks of sound-only therapy. This outcome is not surprising given the history of sound therapy. The study clearly demonstrates that people with more severe self-perceived tinnitus handicap are more likely to get benefit from bimodal stimulation, but the manuscript does not emphasize the fact that people can respond to acoustic-only stimulation, which seems misleading.

The first sentence of the manuscript is misleading; tinnitus is estimated to be present in about 10-15% of the population, but most people with tinnitus are not negatively affected.

It would be interesting to know if there were differences in hearing level between responders and nonresponders. Is there any spectral adjustment to the acoustic stimulation for participants with hearing loss to account for hearing loss configuration?

After reading and re-reading, it is my understanding that all participants (n=112) had baseline THI scores > 38, but after 6 weeks of sound only treatment, only n=44 still had THI scores > 38. This is not clearly indicated. Please clarify in the manuscript (e.g., Figure 1) between the n=112 and the n=44 (i.e., Figure 1).

The first sentence of the discussion suggests that bimodal stimulation is critical for any improvement, which is not the outcome of the study. The criticality of sound and tongue stimulation components applies to a subpopulation of patients with tinnitus – those who do not respond sufficiently to sound alone. This is mentioned near the end of the Discussion but would be appropriate to mention sooner.

Please consider discussing the difference between 79.4% who had a clinically significant change in the THI and only 62.9% who subjectively noted improvement.

Reviewer #2 (Remarks to the Author):

Authors conducted a clinical trial to evaluate the efficacy of a device named Lenire developed by Neuromod Devices, Ireland for a novel treatment of tinnitus. The device allows for application of electrical tongue stimulation along with sound stimulus. Earlier studies in animal models have shown that such a bimodal treatment is one of the strongest drivers of neuroplasticity within the auditory cortex and midbrain. They designed a sound protocol based on two earlier large-scale clinical trials across 517 tinnitus patients which focused on parameter optimization. The paper has been written with clarity and the results have been provided with sufficient context and consideration of previous work.

The study is significant since it involves translational research involving this novel approach which looked at obtaining superior performance of bimodal treatment compared to sound therapy alone. By USA FDA standards the outcomes were sufficiently positive and resulted in FDA De Novo approval.

Quality of data is technically sound and analyzed thoroughly resulting in rigorous interpretation. The authors conducted a clinical trial based on USA FDA guidance and evidence gathered from two previous clinical trials that used Lenire device for bimodal neuromodulation involving electrical tongue stimulation besides sound. During the 12 week study period there were no severely adverse events. The study design was sound and lead to United States FDA approval as a new medical device category. However, the fact that the male to female ratio of the subjects was 3:1

indicates a bias in subject pool.

Supplementary documentation is very helpful in assessing the soundness of methods/protocol used as well as strong evidence for author's claims.

It is questionable if the results of the study would have a major impact on advancing the field of bimodal neuromodulation. US FDA focuses more on lack of harmfulness of the new treatment during clinical trials and efficacy of equal or better than standard care is acceptable. However, for scientific impact of using electrical tongue stimulation along with sound therapy results presented in this work are not very encouraging.

The authors claim that the performance was superior for 112 participants based on responder rate analysis involving participants' recommendation to other patients. However this is subjective and results of validated outcome instruments like Tinnitus Handicap inventory (THI) and Tinnitus Functional Index (TFI) are critical in determining the impact of the treatment. The authors report that responder rate based on THI was 43.3 % for those who underwent the bimodal neuromodulation treatment. On the other hand purely sound treatment resulted in responder rate of 63.3%. This result is not likely to have an impact on the field of bimodal neuromodulation based treatments. Scientifically the result of the study would have had significance if the responder rate was greater after the treatment. Even though the number of subjects enrolled (112) was sufficient for USA FDA to approve the device, scientifically it is a very low number and a significantly larger number of subjects (500 or more) should be studied. Thus the comprehensiveness of statistical basis is weak. Results reported by two earlier scientific studies on 517 tinnitus patients did provide strong and consistent evidence that bimodal stimulation involving sound and electrical tongue stimulation can reduce tinnitus symptoms.

The authors report that for 44 subjects the THI was more than 38 (moderate to severe tinnitus) after the sound treatment and they showed clinically significant superior performance. It is recommended that authors conduct studies targeted to this patient population. As noted before 44 is a very small number and for scientific significance a larger pool of subjects should be targeted as in case of two earlier trials (517). It would really help to lengthen the study period (follow up after the treatment) to determine how long the tinnitus symptoms remain suppressed by the proposed treatment and if it is permanent.

Previous studies included both THI and TFI as outcome instruments. It is not clear if this study included TFI scores. Authors have not reported any results for TFI and inclusion of this additional instrument would improve the scientific understanding. No justification is provided for not addressing TFI. If the results were poor with regards to TFI they still need to be reported for a better understanding of the scientific study.

Reviewer #3 (Remarks to the Author):

A large number of participants were screened by excluded. Reasons for exclusion should be given along with numbers in figure 1.

"As shown in Table 1, there were no significant differences between sites pertaining to several demographic characteristics, except for THI scores for the screened participants in which the Gent site was slightly lower than the other two clinical sites."

Are demographic differences between sites important? Even if so, I wouldn't expect them to be in Table 1, or compared statistically. The aim of the study is not to compare sites, and it's not powered to do so, so apparent lack of difference could just be type II error. Comparisons between sites should be mentioned in the methods if it is retained, and probably relegated to supplementary results.

Responder rates are written as e.g. 43.3% ± 4.79%. It would be better to write as estimate (95% CI: x to y).

I'm confused by responder being defined as "a participant with an improvement in THI score of at least 7 points", but then responders at stage 2 described as "The 43.3% response during Stage 2 is the additional clinically significant improvement in tinnitus symptoms (at least 7 more points in

THI) when adding tongue stimulation to sound-only stimulation above what was already achieved during Stage 1 with sound-only stimulation of 63.3%". To be a responder at stage 2, do they need to improve by a further 7 points on top of 7 points from stage 1? This might be clearer if number of responders and denominators are given instead of just percentages (also in Table 2).

On the face of it, the response rate is lower in the at stage 2, so the additional tongue stimulation seems to make the intervention less effective.

A major limitation of this study that doesn't seem to be discussed is that all participants received stage 1 and stage 2 in the same order. It could be that the addition of tongue stimulation at stage 2 has no effect, and increased effects after stage 2 could just be due to the continued sound therapy. It seems to me that a RCT with sound therapy as the control and sound therapy + tongue stimulation as the intervention (potentially with crossover) would have been a more informative design.

Statistical analysis

What was the justification for using a one-sided test?

What's the justification for the primary analysis method? It seems to me that the appropriate method would have been mixed effects logistic regression, with responder (Yes/No) as the outcome, a random effect for participant to account for pairing and a fixed effect for stage to compare stage 2 to stage 1. Adjustments for any variables thought to be related to the outcome could be made if any are thought to be important (they were important enough to compare between sites).

Where's the sample size calculation? This must be provided.

Nature Communications Manuscript -- Response to Reviewers:

Reviewer #1 (Remarks to the Author):

--This manuscript describes a follow up study of a bimodal stimulation device, the Lenire, designed to decrease tinnitus symptoms. The manuscript is very well written. Many thanks to the authors for a very interesting study. The results are noteworthy in that a significant proportion of the participants demonstrated reductions in self-perceived tinnitus handicap following 6 weeks of sound-only therapy. This outcome is not surprising given the history of sound therapy. The study clearly demonstrates that people with more severe self-perceived tinnitus handicap are more likely to get benefit from bimodal stimulation, but the manuscript does not emphasize the fact that people can respond to acoustic-only stimulation, which seems misleading.

>>We greatly thank the reviewer for such positive comments on the study and in indicating it was a very well written paper, as well as for taking the time to complete such a detailed review of our study. We hope we have suitably addressed all queries and comments below.

The reviewer correctly points out that those with more severe tinnitus (i.e., THI ≥ 38) are expected to obtain greater benefit from bimodal stimulation, as well as people also being able to benefit from acoustic-only stimulation. Our intention was not to indicate that sound therapy is not effective. Our main intention with the primary endpoint was to confirm that bimodal stimulation achieves greater improvement than sound-only stimulation, which was confirmed in those with moderate or more severe tinnitus. To address the reviewer's concern, we have better emphasized the point about positive benefit and use of sound therapy in the final paragraph of the Discussion.

"Another consideration for Lenire bimodal treatment is that clinical efficacy was specifically demonstrated for individuals who have moderate or worse tinnitus symptoms (i.e., THI score ≥ 38). For those with mild or slight tinnitus symptoms (i.e., THI < 38), and considering the positive benefit with sound-only stimulation observed in Stage 1, sound therapy can be initially considered for these tinnitus patients and if symptoms are not sufficiently addressed, then bimodal treatment may also be provided as a follow-on therapy. Encouragingly, 64.9% of participants who did not clinically improve in response to sound therapy during Stage 1 of the study exhibited clinically meaningful improvements in tinnitus symptoms in response to bimodal stimulation during Stage 2. Therefore, Lenire bimodal treatment offers a clinically validated approach that is CE marked in Europe and now FDA approved in the United States that can be used to treat tinnitus patients as a standalone approach or in combination with various sound therapies already available in hearing clinics."

--The first sentence of the manuscript is misleading; tinnitus is estimated to be present in about 10-15% of the population, but most people with tinnitus are not negatively affected.

>>The sentence has been better clarified with additional references and now reads: *"Tinnitus is a phantom auditory sensation that is coded within the brain that affects 10-15% of the population, where approximately 6-11% of the tinnitus population experience bothersome tinnitus and 2-8% experience a severe form of tinnitus, with limited treatment options."*

--It would be interesting to know if there were differences in hearing level between responders and non-responders. Is there any spectral adjustment to the acoustic stimulation for participants with hearing loss to account for hearing loss configuration?

>>This is an interesting question brought up by the reviewer about hearing level effects on outcomes. We did not pre-specify this type of analysis in the study, so we are cautious to make any claims in that direction and we do not feel we have sufficient data to fully answer that question. At least based on mean hearing loss as defined for Table 1, we did not see any differences in responder rates; however, there are various ways we can define hearing loss, so we are interested to pursue this intriguing question for future studies and analyses. In response to the spectral adjustment question, the acoustic stimulation was compensated to the hearing loss for each participant; this is described in the Methods section on page 26.

“The participant’s pure-tone audiometric threshold (250 Hz to 8 kHz) was measured at the screening visit and subsequently used to configure the sound stimuli to be comfortably audible above their hearing threshold at each tone frequency.”

--After reading and re-reading, it is my understanding that all participants (n=112) had baseline THI scores > 38, but after 6 weeks of sound only treatment, only n=44 still had THI scores > 38. This is not clearly indicated. Please clarify in the manuscript (e.g., Figure 1) between the n=112 and the n=44 (i.e., Figure 1).

>>We agree this point can be confusing. The reviewer correctly understood that there is n=44 in the ≥ 38 THI cohort from the TENT-A3 study; this cohort was further supported by extensive real-world evidence in 276 patients who were seeking help for their tinnitus in a clinic where the data were submitted for FDA approval and also included in the paper. We have clarified the n for this cohort in the Figure 1 Legend: *“Intention-to-Treat (ITT) population consists of all participants who met the eligibility criteria (including a THI ≥ 38 at the screening visit), were enrolled in the investigation, and were fitted with the investigational device. The ITT full cohort (n=112) consists of all participants from the enrolment visit, while the ITT moderate or worse severity cohort (n=44) consists of participants who had a THI ≥ 38 at the 6-week interim visit.”*

--The first sentence of the discussion suggests that bimodal stimulation is critical for any improvement, which is not the outcome of the study. The criticality of sound and tongue stimulation components applies to a subpopulation of patients with tinnitus – those who do not respond sufficiently to sound alone. This is mentioned near the end of the Discussion but would be appropriate to mention sooner.

>>We would like to clarify that the study results demonstrate the criticality of sound and tongue stimulation components for those who have tinnitus that is moderate or more severe (i.e., THI ≥ 38 when starting bimodal stimulation), regardless if they have sufficiently benefited to acoustic-only stimulation or not. We have further clarified this point in the first paragraph of the Discussion and also in referring to Table 2 and Figure 2.

“For individuals with a moderate to catastrophic severity of tinnitus handicap when starting bimodal treatment (i.e., THI score ≥ 38), a clinically significant superior performance of the Lenire device with combined sound and tongue stimulation was achieved with 6 weeks of treatment (58.6%) when compared to sound-only stimulation (43.2%; $p=0.022$; Table 2). This outcome occurred regardless of whether the participant benefitted or not from sound-only stimulation (Fig. 2).”

--Please consider discussing the difference between 79.4% who had a clinically significant change in the THI and only 62.9% who subjectively noted improvement.

>>To clarify, the 79.4% responder rate pertains to the full 12-weeks of treatment while the satisfaction rate of 62.9% refers to the benefit of the device during 6-weeks of bimodal stimulation during the second stage of treatment. This has been clarified in the first paragraph of the Discussion.

“At the final visit, 62.9% of participants indicated that they benefited from using the Lenire device with bimodal stimulation, which corresponds to the 6-week treatment period of Stage 2. Also, 88.6% said that if they knew someone with tinnitus, they would recommend they try this treatment.”

Reviewer #2 (Remarks to the Author):

--Authors conducted a clinical trial to evaluate the efficacy of a device named Lenire developed by Neuromod Devices, Ireland for a novel treatment of tinnitus. The device allows for application of electrical tongue stimulation along with sound stimulus. Earlier studies in animal models have shown that such a bimodal treatment is one of the strongest drivers of neuroplasticity within the auditory cortex and midbrain. They designed a sound protocol based on two earlier large-scale clinical trials across 517 tinnitus patients which focused on parameter optimization. The paper has been written with clarity and the results have been provided with sufficient context and consideration of previous work.

The study is significant since it involves translational research involving this novel approach which looked at obtaining superior performance of bimodal treatment compared to sound therapy alone. By USA FDA standards the outcomes were sufficiently positive and resulted in FDA De Novo approval.

Quality of data is technically sound and analyzed thoroughly resulting in rigorous interpretation. The authors conducted a clinical trial based on USA FDA guidance and evidence gathered from two previous clinical trials that used Lenire device for bimodal neuromodulation involving electrical tongue stimulation besides sound. During the 12 week study period there were no severely adverse events. The study design was sound and lead to United States FDA approval as a new medical device category.

>>We greatly appreciate the recognition by the reviewer that the study and data analysis were performed and presented in a rigorous and sound manner and that the paper was written in a clear way with sufficient context and background for interpretation; with all of this being sufficiently positive to lead to FDA De Novo approval as a new medical device category. We also greatly appreciate the reviewer’s time and effort to review our manuscript. We hope we have suitably addressed all queries and comments below.

--However, the fact that the male to female ratio of the subjects was 3:1 indicates a bias in subject pool.

>>The reviewer is correct that we had a higher ratio of males to females at about 70% to 30% in our study. This bias in subject pool of males to females is in line with the real-world situation, where the prevalence of tinnitus is higher in males than females across demographic groups (Batts et al.,2024, Hackenberg et al.,2023, McCormack et al., 2016). Males are also more likely than females to discuss tinnitus with a healthcare provider (Bhatt 2016). Since this study was designed in order to obtain FDA approval, generalizability to the overall tinnitus population seeking clinical care in the real-world was important to FDA and deemed sufficient in terms of granting approval for Lenire treatment. We have clarified a few of these points in the third paragraph of the Results section, as well as

addressing the editor's point to provide primary analysis based on sex, which is now included as Supplementary Table 2. The real world evidence included in the paper also shows that those seeking Lenire treatment consist of more males than females (see Supplementary Table 3).

--Supplementary documentation is very helpful in assessing the soundness of methods/protocol used as well as strong evidence for author's claims. It is questionable if the results of the study would have a major impact on advancing the field of bimodal neuromodulation. US FDA focuses more on lack of harmfulness of the new treatment during clinical trials and efficacy of equal or better than standard care is acceptable. However, for scientific impact of using electrical tongue stimulation along with sound therapy results presented in this work are not very encouraging.

>>We really appreciate the acknowledgement of our continued efforts to be transparent and rigorous by sharing our detailed study protocols and results alongside the supplementary documents included for the publication, which allowed the reviewers to assess the soundness of the study design and strong evidence for our claims in the paper. We agree with the reviewer that this study does not contribute in a major way to the science side of the bimodal neuromodulation field as was achieved by previous published studies by several of the co-authors of this current paper. The current paper and TENT-A3 study has been more focused on advancing the clinical application of bimodal neuromodulation for real-world impact. We appreciate the reviewer acknowledging that the clinical outcomes, showing the safety and efficacy of our device, were acceptable to the FDA to approve the device as having a clinically significantly superior performance compared to standard of care for the moderate and above severity cohort. For the purpose of advancing the clinical side of bimodal neuromodulation, we also included extensive real-world evidence involving 276 patients who were seeking help for their tinnitus in a clinic where the data were also submitted to FDA and included in the paper; those results were quite consistent with the TENT-A3 study results, which FDA viewed as convincing real world evidence for clinical implementation and benefit to tinnitus patients to grant De Novo approval.

--The authors claim that the performance was superior for 112 participants based on responder rate analysis involving participants' recommendation to other patients. However this is subjective and results of validated outcome instruments like Tinnitus Handicap inventory (THI) and Tinnitus Functional Index (TFI) are critical in determining the impact of the treatment.

>>We fully agree with the reviewer that validated outcome instruments, such as THI, are critical to determining impact of the treatment and is what we used for the responder rate analysis referred to by the reviewer. To better clarify this point, we emphasized THI usage by adding it to the title of Table 2 so there is no confusion what was used for our primary endpoint analyses.

Title updated: *"Table 2: Primary endpoint analyses for bimodal treatment for tinnitus based on THI"*

--The authors report that responder rate based on THI was 43.3 % for those who underwent the bimodal neuromodulation treatment. On the other hand purely sound treatment resulted in responder rate of 63.3%. This result is not likely to have an impact on the field of bimodal neuromodulation based treatments. Scientifically the result of the study would have had significance if the responder rate was greater after the treatment. Even though the number of subjects enrolled (112) was sufficient for USA FDA to approve the device, scientifically it is a very low number and a significantly larger number of subjects (500 or more) should be studied. Thus the comprehensiveness of statistical basis is weak. Results reported by two earlier scientific studies on 517 tinnitus patients did provide strong and consistent evidence that bimodal stimulation involving

sound and electrical tongue stimulation can reduce tinnitus symptoms. The authors report that for 44 subjects the THI was more than 38 (moderate to severe tinnitus) after the sound treatment and they showed clinically significant superior performance. It is recommended that authors conduct studies targeted to this patient population. As noted before 44 is a very small number and for scientific significance a larger pool of subjects should be targeted as in case of two earlier trials (517).

>>The reviewer correctly points out that the two previous studies on Lenire bimodal treatment were significantly large and provided strong/consistent evidence for using sound and electrical tongue stimulation to treat tinnitus. For FDA approval in confirming clinical safety and efficacy, the participant numbers used for the TENT-A3 study were sufficient, as the reviewer also correctly stated. It should be further noted that we had to obtain FDA “De Novo” approval for a completely new medical device category that we created for bimodal neuromodulation, so the success criteria were even higher for demonstrating clinical safety and efficacy than a traditional 510(k) regulatory pathway as is used for multiple sound and hearing aid approaches to treat tinnitus. We acknowledge the reviewer’s comment about confirming clinically significant superior performance over sound treatment in moderate and more severe tinnitus patients (based on THI) in 44 subjects, and scientifically it would have been helpful to have larger numbers of subjects. Fortunately, we also included extensive real-world evidence in 276 patients who were seeking help for their tinnitus in a clinic, which was submitted to FDA and included in the paper; those results were quite consistent with the efficacy outcomes from those 44 subjects confirming clinically significant superior performance over sound treatment in moderate and more severe tinnitus patients based on THI. Therefore, the totality of data presented in the paper was based on quite a large number of subjects, which FDA viewed as convincing evidence for clinical implementation and benefit to tinnitus patients to grant De Novo approval.

--It would really help to lengthen the study period (follow up after the treatment) to determine how long the tinnitus symptoms remain suppressed by the proposed treatment and if it is permanent.

>>We also wished we could have performed a follow-up period in the TENT-A3 study, as was done in the previously published TENT-A1 and TENT-A2 studies. Due to the COVID pandemic restrictions and challenges, along with the short time period available to complete this study within our FDA De Novo timeline requirements, we were not able to add a lengthy follow-up period as was done in those previous studies. In the TENT-A1 and TENT-A2 studies, which enrolled over 500 participants and published results in rigorous peer-reviewed publications (i.e., in Science Translational Medicine and Nature Scientific Reports), tinnitus individuals were tracked after treatment stopped, in which it was consistently shown across both studies that tinnitus symptoms could be improved up to 12 months after Lenire treatment. The ability to obtain consistent evidence for the Lenire treatment across two separate large-scale clinical trials showing this long-term effect, gives no reason to believe that such long-term benefit would not have been observed in this TENT-A3 FDA trial. This long-term benefit consideration has been discussed in the fourth paragraph of the Discussion since we also viewed this as an important point to bring up in the paper.

--Previous studies included both THI and TFI as outcome instruments. It is not clear if this study included TFI scores. Authors have not reported any results for TFI and inclusion of this additional instrument would improve the scientific understanding. No justification is provided for not addressing TFI. If the results were poor with regards to TFI they still need to be reported for a better understanding of the scientific study.

>>TFI was not originally included in this paper. Both THI and TFI were co-primary endpoints in the previous TENT-A1 and TENT-A2 studies, and thus those results were presented in their corresponding publications. The primary endpoint for the TENT-A3 study and sample size powering of the study were based solely on THI and not TFI, based on the study that we designed with FDA. As was demonstrated in the previous large-scale TENT-A1 and TENT-A2 studies, the THI and TFI results have been consistent with each other in demonstrating clinical efficacy of Lenire bimodal treatment, and both are also consistent in the current TENT-A3 study (e.g., for TFI the addition of tongue stimulation to sound therapy had a responder rate of 45.5% versus only 29.6% for sound therapy alone, which was a significantly better p-value of 0.010 than observed for THI; responder defined as 13 points change in TFI and for those with moderate or more severe tinnitus symptoms). We did not originally include TFI in this paper because we intended to focus on the primary endpoint analysis and key results based on THI that were proposed to FDA for our study. We then planned to present those other exploratory analyses, including for TFI and HUI3, in a follow-on publication that we are currently preparing. The last sentences of the Introduction of the originally submitted paper included text to explain this point: *“Primary endpoint analyses, satisfaction outcomes, and safety results are presented in this paper. Other planned analyses, including treatment effects in different subgroups of tinnitus patients and with alternative outcome instruments, will be presented in a subsequent publication.”*

To fully address the reviewer’s comment, we included TFI results in the paper (page 13 of revised paper) to allow direct comparison with the THI results. We also included HUI3 results (page 18), which was another exploratory endpoint measure used in the TENT-A3 study. We have modified the last sentences of the Introduction, as well as the Abstract and Methods, to reflect these updates.

Reviewer #3 (Remarks to the Author):

--A large number of participants were screened by excluded. Reasons for exclusion should be given along with numbers in figure 1.

>>We appreciate the feedback from the reviewer. We have included additional information relating to the excluded participants. We performed pre-screening online to handle the large number of interested participants in the study; those data are not sufficiently available for the purpose requested by the reviewer. We are able to provide the reasons for exclusion for those participants who were screened in the clinics (222 total), in which 110 participants were not enrolled in the study. 11 of the 110 participants met the eligibility criteria and were put on a waiting list who were not enrolled. Of the remaining participants, 95 were screening failures, while 4 did not come to the ENROLMENT visit. The dominant reasons for screening failure were due to a baseline THI lower than 38 (i.e., less than moderate severity level; 34.7%, 33/95); hearing loss of greater than 80dB HL in any test frequency in the set {2k,3k,4k,6k,8k} Hz or greater than 40 dB HL in the set {250,500,1k} Hz either unilaterally or bilaterally (22.1%, 21/95); or previous involvement in a clinical investigation for tinnitus or had an experimental/surgical treatment for tinnitus (20.0%, 19/95). We have added a table with the screen failure reasons for those 110 participants not enrolled in the study in Supplementary Table 1 and refer to it in the first paragraph of the Results section.

--As shown in Table 1, there were no significant differences between sites pertaining to several demographic characteristics, except for THI scores for the screened participants in which the Gent site was slightly lower than the other two clinical sites.” Are demographic differences between sites important? Even if so, I wouldn’t expect them to be in Table 1, or compared statistically. The aim of

the study is not to compare sites, and it's not powered to do so, so apparent lack of difference could just be type II error. Comparisons between sites should be mentioned in the methods if it is retained, and probably relegated to supplementary results.

>> Based on the reviewer's correct comment, we have removed the comparison across sites and now only provide the data for the Full Cohort in Table 1.

--Responder rates are written as e.g. 43.3% ± 4.79%. It would be better to write as estimate (95% CI: x to y).

>>We have updated the format as requested throughout the paper where we also include the 95% CI as well whenever presenting the responder rates for the primary endpoint analysis in the main paper.

--I'm confused by responder being defined as "a participant with an improvement in THI score of at least 7 points", but then responders at stage 2 described as "The 43.3% response during Stage 2 is the additional clinically significant improvement in tinnitus symptoms (at least 7 more points in THI) when adding tongue stimulation to sound-only stimulation above what was already achieved during Stage 1 with sound-only stimulation of 63.3%". To be a responder at stage 2, do they need to improve by a further 7 points on top of 7 points from stage 1? This might be clearer if number of responders and denominators are given instead of just percentages (also in Table 2). On the face of it, the response rate is lower in the at stage 2, so the additional tongue stimulation seems to make the intervention less effective.

>>We can understand the confusion and tried to better clarify the criteria in the paper. Through discussions with FDA and based on their recommendations, we used a criteria that was quite a high bar in that we not only had to show bimodal treatment is efficacious, but we had to show it exceeds what is possible with sound-only stimulation in the same participants after they already received sound therapy (i.e., the primary endpoint looks at the responder rate in Stage 2 and compares it to the responder rate in Stage 1). So, a "responder" could respond in both Stage 1 and Stage 2, meaning they would have achieved an improvement by a further 7 points or more in Stage 2 on top of the 7 points or more they achieved in Stage 1, as the reviewer correctly stated. A responder could also have over 7 points improvement in Stage 2 without responding in Stage 1. We have better clarified this criteria on page 12 of the Results section by adding the clarifying text below:

"The 43.3% response during Stage 2 is the additional clinically significant improvement in tinnitus symptoms (at least 7 more points in THI) when adding tongue stimulation to sound-only stimulation above what was already achieved during Stage 1 with sound-only stimulation. In other words, during Stage 1, there were 63.3% of participants who obtained at least 7 points improvement in THI in response to sound-only stimulation with 36.7% of participants obtaining less than 7 points improvement; then in Stage 2, these same participants had to also achieve at least 7 more points of improvement above what they already achieved in Stage 1 to be considered a responder in Stage 2, which was a rigorously high criterion for success."

--A major limitation of this study that doesn't seem to be discussed is that all participants received stage 1 and stage 2 in the same order. It could be that the addition of tongue stimulation at stage 2 has no effect, and increased effects after stage 2 could just be due to the continued sound therapy. It seems to me that a RCT with sound therapy as the control and sound therapy + tongue stimulation as the intervention (potentially with crossover) would have been a more informative design.

>>The design of the study was extensively discussed and evaluated together with FDA. Since the treatment cannot be blinded because both the sound and tongue stimuli are suprathreshold/noticeable where a sham control is not possible, a traditional RCT with crossover design was not feasible. Therefore, FDA requested a rigorously high criterion for success where we had to demonstrate not just that bimodal treatment is effective (e.g., over a sham control), but also that bimodal treatment is effective above and beyond sound-only treatment. Furthermore, the design of the study requires not only that there is additional improvements in Stage 2 with bimodal stimulation, but that there are additional improvements on top of improvements that already happened in Stage 1 with sound-only stimulation. Since the success criterion requires a significantly higher responder rate in Stage 2 compared to the responder rate observed in Stage 1, we would have failed if there was less than a doubling of response rate across Stage 1 through to Stage 2; we actually had to obtain a growing slope of improvement over time that was more than a doubling effect. We have not observed publications, to the best of our knowledge, with appropriately designed sound therapy studies that can achieve a growing slope of improvement over time as treatment continues (e.g., if sound therapy causes more than 7 points of improvement in X participants during 6 weeks, then sound therapy needs to cause more than 7 additional points of improvement in X+Y participants during another 6 weeks, where the extra Y participants in Stage 2 is required to reach statistical significance in Stage 2 over Stage 1). Therefore, this success criterion is quite rigorously high to show bimodal treatment is effective and above sound-only stimulation effects that the FDA urged us to pursue for De Novo approval, which fortunately we successfully confirmed for participants with moderate or more severe tinnitus.

To clarify these points, we added another paragraph in the Discussion section (second to last paragraph of that section), which is provided below:

“One consideration for interpreting the results of bimodal treatment is related to the design of the study. The Lenire bimodal treatment cannot be sufficiently blinded because both the sound and tongue stimuli are suprathreshold and noticeable, where a sham control is not possible. Therefore, a high criterion for efficacy success was decided together with FDA, where we had to demonstrate not only that bimodal treatment is effective (e.g., over a sham control), but also that bimodal treatment is more effective than sound-only treatment. Furthermore, the design of the study required that additional improvements in tinnitus symptoms is achieved with bimodal stimulation in Stage 2 above and beyond the improvements already achieved in Stage 1 with sound-only stimulation. In other words, the success criterion requires more than a doubling of response rate across Stage 1 through to Stage 2. We have not observed, to the best of our knowledge, appropriately designed sound therapy studies demonstrating a growing slope of improvement over time as treatment continues. Overall, our rigorously designed success criterion confirms that Lenire bimodal treatment achieves clinical efficacy for participants with moderate or worse tinnitus.”

--Statistical analysis: What was the justification for using a one-sided test?

>>A single-sided hypothesis was based on previously published TENT-A1 and TENT-A2 results supporting that one direction of effect is clinically relevant. To ensure an appropriately rigorous statistical criterion, and in discussion with the FDA, the significance level was accordingly adjusted to 0.025 (in lieu of using a significance level of 0.05 and a two-sided hypothesis). The net effect of this adjustment to the significance level of 0.025 is comparable to the p-value in the two-sided case of 0.05; and thus, the directionality of the hypothesis did not impact our conclusions of the study. We clarified these points in the first paragraph of the Statistical Analyses section of the Methods.

--What's the justification for the primary analysis method? It seems to me that the appropriate method would have been mixed effects logistic regression, with responder (Yes/No) as the outcome, a random effect for participant to account for pairing and a fixed effect for stage to compare stage 2 to stage 1. Adjustments for any variables thought to be related to the outcome could be made if any are thought to be important (they were important enough to compare between sites).

>>For our single-arm repeated measures study design, and based on discussions with our CRO (Avania), statistical consultants, and FDA biostatisticians, an appropriate primary analysis method was determined to be a single sample, one-sided normal approximation test (Z-test) for a binomial proportion with a significance level of 0.025 in order to test the responder rate in the second 6-weeks, attributed to the addition of tongue stimulation to sound-only stimulation, compared to the point-estimate of the responder rate in the first 6-weeks for sound-only stimulation. This is the test that was specifically pre-specified for the primary study analysis and approved by FDA; thus, it is the test that was required for the study and for obtaining FDA De Novo approval. We included a sentence at the start of the Results-Statistical Analyses section to clarify this point: *"The primary endpoint analysis was developed together with the CRO Avania and with guidance from the FDA that was viewed as appropriate for our study design and Lenire treatment considerations."*

--Where's the sample size calculation? This must be provided.

>>The sample size calculation was originally included in the attached documents that were included with the manuscript as part of the Supplementary Information (i.e., Clinical Investigation Plan and Statistical Analysis Plan). In response to the reviewer's request, we have included it also in the Methods at the end of the Statistical Analyses section.

REVIEWERS' COMMENTS

Reviewer #1 (Remarks to the Author):

The authors have adequately addressed this reviewer's concerns.

Reviewer #2 (Remarks to the Author):

Accept the changes made in the revised manuscript. I am satisfied with the rebuttal

Reviewer #3 (Remarks to the Author):

My primary concerns with the previous version of the paper were around clarity, as some parts were quite complex. The authors have addressed my concerns sufficiently. It sounds like the important design and analysis of this study were discussed extensively, which is reassuring.